# Cooling a mechanical resonator with nitrogen-vacancy centres using a room temperature excited state spin–strain interaction

E.R. MacQuarrie[1], M. Otten[1], S.K. Gray[2] & G.D. Fuchs[3]

Cooling a mechanical resonator mode to a sub-thermal state has been a long-standing challenge in physics. This pursuit has recently found traction in the field of optomechanics in which a mechanical mode is coupled to an optical cavity. An alternate method is to couple the resonator to a well-controlled two-level system. Here we propose a protocol to dissipatively cool a room temperature mechanical resonator using a nitrogen-vacancy centre ensemble. The spin ensemble is coupled to the resonator through its orbitally-averaged excited state, which has a spin–strain interaction that has not been previously studied. We experimentally demonstrate that the spin–strain coupling in the excited state is $13.5 \pm 0.5$ times stronger than the ground state spin–strain coupling. We then theoretically show that this interaction, combined with a high-density spin ensemble, enables the cooling of a mechanical resonator from room temperature to a fraction of its thermal phonon occupancy.

[1] Department of Physics, Cornell University, Ithaca, New York 14853, USA. [2] Center for Nanoscale Materials, Argonne National Laboratory, Argonne, Illinois 60439, USA. [3] School of Applied and Engineering Physics, Cornell University, Ithaca, New York 14853, USA. Correspondence and requests for materials should be addressed to G.D.F. (email: gdf9@cornell.edu).

ooling a mechanical resonator to a sub-thermal phonon occupation can enhance sensing by lowering the resonator's thermal noise floor and extending a sensor's linear dynamic range[1–4]. Taken to the extreme, cooling a mechanical mode to the ground state of its motion enables the exploration of quantum effects at the mesoscopic scale[5–7]. These goals have motivated researchers in the field of optomechanics to invent methods for cooling mechanical resonators through their interactions with light. Such techniques have been able to achieve cooling to the ground state from cryogenic starting temperatures[6,7] and to near the ground state from room temperature[8–12].

A well-controlled quantum system coupled to the motion of a resonator can also be used to cool a mechanical mode[13,14]. Recently, nitrogen-vacancy (NV) centres in diamond have been coupled to mechanical resonators through coherent interactions with lattice strain[15–22]. The opportunity to use these interactions has stimulated the development of single-crystal diamond mechanical resonators[23–26] and motivated several theoretical proposals for cooling such resonators with a single NV centre[13,14,27,28]. In principle, replacing the single NV centre with a many-NV ensemble can provide a collective enhancement to the strain coupling, which could increase the cooling power of these protocols. In practice, however, ensembles can shorten spin coherence times and introduce inhomogeneities that may make collective enhancement impractical, depending on the proposed mechanism. To make ensemble coupling a useful resource, it thus becomes crucial to design a cooling protocol that is insensitive to these side effects.

In this work, we study the hybrid quantum system composed of an NV centre spin ensemble collectively coupled to a mechanical resonator with the goal of developing a method for cooling the resonator from ambient temperature. Experimentally, we characterize the previously unstudied spin–strain coupling within the room temperature NV centre excited state (ES), and we find that it is $13.5 \pm 0.5$ times stronger than the ground state (GS) spin–strain interaction. We then propose a dissipative cooling protocol that uses this ES spin–strain interaction and theoretically show that a dense NV centre ensemble can cool a high-$Q$ mechanical resonator from room temperature to a fraction of its thermal phonon population. The proposed protocol requires neither long spin coherence times nor strong spin-phonon coupling, and the cooling power scales directly with the NV centre density. These properties make our proposed protocol a practical approach to cooling a room temperature resonator.

## Results

**NV centre–strain interactions**. To achieve substantial cooling from ambient conditions, we require a room temperature NV centre–strain interaction that can be enhanced by an ensemble. We first consider the orbital-strain coupling that exists within the NV centre ES at cryogenic temperatures. This $850 \pm 130$ THz per strain interaction offers a promising route towards single NV centre-mechanical resonator hybrid quantum systems[21,22]. For ensemble coupling, however, inevitable static strain inhomogeneities will strongly broaden the orbital transition and prohibit collective enhancement. Moreover, the orbital coherence begins to dephase above 10 K because of phonon interactions[29], limiting applications of orbital-strain coupling to cryogenic operation.

A weaker ($21.5 \pm 1.2$ GHz per strain) spin–strain coupling exists at room temperature within the NV centre GS (ref. 17). The resonance condition for this interaction is determined by a static magnetic bias field which can be very uniform across an ensemble. This GS spin–strain interaction thus offers a path towards coupling an ensemble to a mechanical resonator. As the NV centre density grows, however, the GS spin coherence will decrease[30,31], limiting the utility of the collective enhancement.

Finally, we consider spin–strain interactions in the room temperature ES, which have not been thoroughly investigated but might provide the desired compatibility with dense ensembles. For temperatures above $\sim 150$ K, orbital-averaging from the dynamic Jahn–Teller effect erases the orbital degree of freedom from the NV centre ES Hamiltonian, resulting in an effective orbital singlet ES at room temperature[29,32–34]. Previously, magnetic spectroscopy measured an unidentified spin splitting within the room temperature ES that is on the order of 10 times stronger than the GS spin–strain interaction. These measurements hinted that this splitting might be a spin–strain interaction in the ES (refs 35,36). Like the GS spin–strain coupling, the resonance condition for such an interaction would be determined by a static magnetic bias field, enabling collective enhancement with an ensemble. Furthermore, the NV centre density is not expected to affect the ES coherence time, which is limited by the ES motional narrowing rate[34,37]. Such an ES spin–strain interaction could thus offer a promising path towards coupling a dense NV centre ensemble to a mechanical resonator. Our first goal then becomes to understand and precisely quantify this coupling.

Assuming this ES coupling is the result of a spin–strain interaction, we can write the spin Hamiltonian for an NV centre in the presence of a magnetic field **B** and non-axial strain $\epsilon_x$. Both the GS and room temperature ES Hamiltonians then take the form ($\hbar = 1$)[36,38]

$$H = D_0 S_z^2 + \gamma_{NV} \mathbf{S} \cdot \mathbf{B} - d_\perp \epsilon_x \left( S_x^2 - S_y^2 \right) + A_{\parallel} S_z I_z \qquad (1)$$

where $D_0^e/2\pi = 1.42$ GHz and $D_0^g/2\pi = 2.87$ GHz are the ES and GS zero-field splittings, $\gamma_{NV}/2\pi = 2.8$ MHz/G is the NV centre gyromagnetic ratio, $A_{\parallel}^e/2\pi = +40$ MHz (ref. 39) and $A_{\parallel}^g/2\pi = -2.166$ MHz are the ES and GS hyperfine couplings to the $^{14}$N nuclear spin, **S** (**I**) is the electronic (nuclear) spin-1 Pauli vector, and the $z$-axis runs along the NV centre symmetry axis. Perpendicular strain $\epsilon_x$ couples the $|(m_s =) +1\rangle$ and $|-1\rangle$ spin states with a strength $d_\perp^e$ in the ES and $d_\perp^g/2\pi = 21.5 \pm 1.2$ GHz per strain in the GS (ref. 17). As shown in Fig. 1a, this interaction enables direct control of the magnetically-forbidden $|+1\rangle \leftrightarrow |-1\rangle$ spin transition within each orbital through resonant strain.

**Device details**. The combination of a large hyperfine splitting in the ES and a short ES lifetime broadens the spectral features of the ES spin–strain interaction. Measuring such a spectrum then requires large magnetic field sweeps ($\Delta B_z \sim 150$ G), which in turn require a mechanical driving field with a high carrier frequency ($\omega_m/2\pi \gtrsim 420$ MHz). To this end, we fabricate a high-overtone bulk acoustic resonator (HBAR) capable of generating large amplitude strain at gigahertz-scale frequencies. The resonator used in this work was driven at a $\omega_m/2\pi = 529$ MHz mechanical mode that has a quality factor of $Q = 1,790 \pm 20$. An antenna fabricated on the opposite diamond face provides high-frequency magnetic fields for magnetic spin control. The final device is pictured in Fig. 1b.

**Spin–strain spectroscopy**. To measure mechanical spin driving within the ES, we execute the pulse sequences shown in Fig. 2a, as a function of the magnetic bias field $B_z$. In the first sequence, a 532 nm laser initializes the NV centre ensemble into the GS level $|g, (m_s =)0\rangle$. A magnetic adiabatic passage (AP) then moves the spin population to $|g, -1\rangle$. At this point, we pulse the mechanical resonator at its resonance frequency $\omega_m$ for 3 μs.

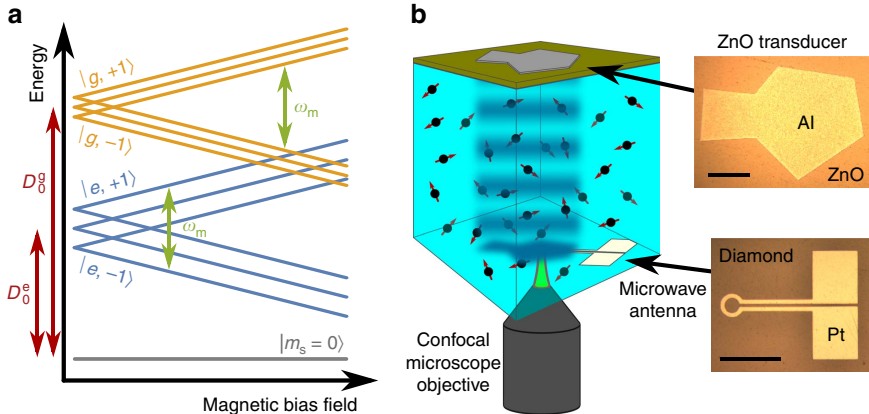

**Figure 1 | Energy levels and device schematic.** (**a**) NV center ground state and excited state energy levels as a function of the magnetic bias field. Energies have been plotted relative to the $m_s = 0$ state in each orbital, and a mechanical mode of frequency $\omega_m$ has been drawn connecting the $m_I = +1$ hyperfine sublevels. (**b**) Schematic of the device used in these measurements along with optical micrographs of the ZnO transducer used to generate the strain standing wave (150 μm scale bar) and the microwave antenna used to generate magnetic control fields (200 μm scale bar).

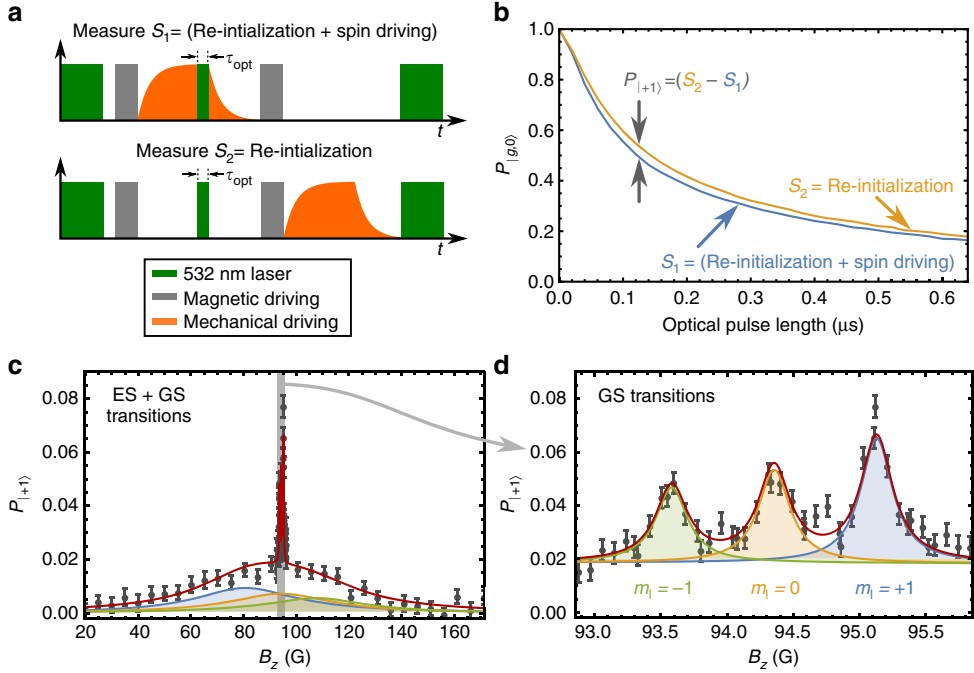

**Figure 2 | Spin–strain spectroscopy.** (**a**) Pulse sequences used to measure excited state (ES) spin driving. (**b**) Population in $|g, 0\rangle$ at the end of the pulse sequences in **a** plotted against $\tau_{opt}$. (**c**) Spectrum of the spin population driven mechanically into $|+1\rangle$ by the ES and ground state (GS) spin–strain interactions. The red curve is a least squares fit to the sum of six Lorentzians. (**d**) Zoomed in view of the GS spin transitions in **c**. The data in **c**,**d** were measured on one device with an NV centre ensemble, and error bars are from the s.d. in photon counting.

Just before the end of the mechanical pulse, we apply a $\tau_{opt} = 125$ ns optical pulse with the 532 nm laser. This excites the ensemble to $|e, -1\rangle$ and allows the spins to interact with the mechanical driving field in the ES. If the driving field is resonant with the $|e, +1\rangle \leftrightarrow |e, -1\rangle$ splitting, population will be driven into $|e, +1\rangle$. The spins then follow either a spin-conserving relaxation down to $|g, \pm 1\rangle$ or a relaxation to the singlet state $|S_1\rangle$ through an intersystem crossing. The former preserves the spin state information, while relaxing to $|S_1\rangle$ re-initializes the state, erases the stored signal and reduces the overall contrast of the measurement. After allowing the ensemble to relax, we apply the second magnetic AP to return the spin population in $|g, -1\rangle$ to $|g, 0\rangle$ and measure the $|g, 0\rangle$ population via fluorescence read out. We define this signal as $S_1$ and plot it as a function of $\tau_{opt}$ in Fig. 2b.

In the second pulse sequence, the mechanical pulse occurs between the second AP and fluorescence read out. Applying the mechanical pulse with the ensemble in $|g, 0\rangle$ maintains the same power load on the device but does not drive spin population. This sequence measures $S_2$, the re-initialization of the ensemble from the $\tau_{opt}$ optical pulse (Fig. 2b). Subtracting $S_2 - S_1$ gives the probability of finding the ensemble in $|+1\rangle$ at the end of the first sequence. A third sequence with a single AP and a fourth with two APs (both with $\tau_{opt} = 0$) normalize the spin contrast at each $B_z$.

Figure 2c shows the resulting experimental signal. The three broad, low peaks correspond to the hyperfine-split $|e, -1\rangle \rightarrow |e, +1\rangle$ transition, providing definitive evidence of a spin–strain interaction within the room temperature ES. Population is also driven into $|g, +1\rangle$ by the GS spin–strain interaction when the

mechanical driving field is resonant with the $|g, +1\rangle \leftrightarrow |g, -1\rangle$ splitting. Figure 2c thus contains both ES and GS spectra. We fit the data to the sum of three low, broad Lorentzians describing the ES spin–strain interaction and three taller, narrower Lorentzians describing the GS interaction (see Methods). Figure 2d highlights the GS driving with a zoomed in view of Fig. 2c about the GS resonances. The GS peaks have larger amplitudes than the ES peaks because the GS interaction acts for the entire duration of the 3 μs mechanical pulse, whereas the ES interaction only acts during the ~125 ns that the spin population is in the ES. Also, the reversed sign of $A_\parallel^e$ relative to $A_\parallel^g$ is consistent with *ab initio* calculations[40] and was confirmed by measurements presented in Supplementary Note 1 that were conditional on the nuclear spin state.

**Quantification of $d_\perp^e$.** To quantify the strength of the ES spin–phonon interaction, we first calibrate the strain amplitude generated by the HBAR by mechanically driving Rabi oscillations within the GS and extracting the GS mechanical Rabi field $\Omega_g$ as a function of the applied power (Fig. 3a). Next, we spectrally isolate the $|e, +1\rangle \leftrightarrow |e, -1\rangle$ transition by fixing $B_z = 80$ G. At this field, the applied strain is on resonance in the ES but off resonance in the GS (Fig. 2c). We then execute a modified version of the pulse sequence described above. Here, we use ~20 ns magnetic π-pulses to address the $|g, 0\rangle \leftrightarrow |g, -1\rangle$ transition and measure both $S_1$ and $S_2$ as a function of $\tau_{opt}$ for each power level applied to the HBAR.

As Fig. 3b shows, taking $S_2 - S_1$ reveals a competition between mechanical driving into $|e, +1\rangle$ and re-initialization into $|0\rangle$ via optical pumping. For nonzero $\tau_{opt}$, the ES mechanical driving field $\Omega_e$ drives spin population from $|e, -1\rangle$ to $|e, +1\rangle$, increasing $P_{|e, +1\rangle}$, but as $\tau_{opt}$ grows, optical pumping re-initializes the ensemble into $|0\rangle$, vacating the $m_s = \{+1, -1\}$ subspace and reducing $P_{|e, +1\rangle}$ to zero. A seven-level master equation model recreates this competition and provides good fits to the data. From these fits, we extract the value of $\Omega_e$. The Methods section includes a detailed description of this model, which was designed to account for inhomogeneities within the NV centre ensemble and for the polarization of the nuclear spin sublevels, among other effects. Plotting $\Omega_e$ against $\Omega_g$ (Fig. 3c) shows that the transverse spin–strain coupling in the ES is $13.5 \pm 0.5$ times stronger than the GS coupling, or $d_\perp^e/2\pi = 290 \pm 20$ GHz per strain.

**Resonator cooling protocol.** With $d_\perp^e$ quantified, we now present a dissipative protocol for cooling a mechanical resonator with an NV centre spin ensemble. In our proposed protocol, a 532 nm laser continuously pumps the phonon sidebands of the ensemble's optical transition, and a gigahertz frequency magnetic field continuously drives the $|g, 0\rangle \leftrightarrow |g, -1\rangle$ spin transition. This generates a steady state population surplus in $|e, -1\rangle$ as compared with $|e, +1\rangle$, enabling the net absorption of phonons by the ensemble. Spontaneous relaxation and subsequent optical pumping continually re-initialize the system, allowing the phonon absorption cycle to continue. Figure 4a summarizes this process.

The dissipative nature of this protocol enables resonator cooling without requiring strong spin-phonon coupling. Here, we define strong coupling as a single-spin cooperativity $\eta = \lambda^2 T_2^*/\gamma n_{th} > 1$, where $\lambda$ is the single spin-single phonon coupling strength, $T_2^*$ is the inhomogeneous spin dephasing time, $\gamma = \omega_m/Q$ is the mechanical dissipation rate and $n_{th} \sim k_B T/\hbar \omega_m$ is the thermal phonon occupancy of the resonator mode[41]. A cooperativity of $\eta > 1$ marks the threshold for coherent interactions between the spin and the mechanical mode. Non-idealities in spin coherence and resonator fabrication have thus far prevented the experimental realization of NV centre cooperativities approaching unity, especially at room temperature. This makes the proposed dissipative protocol a practical and attractive approach because it does not require coherent interactions for resonator cooling to occur.

To analyse the performance of the protocol we start by considering a single two-state spin system coupled to a mechanical resonator. The resulting dynamics obey the master equation ($\hbar = 1$)

$$\dot\rho = -i[H, \rho] + \mathcal{L}_\Gamma \rho + \mathcal{L}_\gamma \rho \qquad (2)$$

where $H$ describes the coherent coupling between the spin and the resonator, $\mathcal{L}_\Gamma$ describes the incoherent spin processes, and $\mathcal{L}_\gamma$ describes the resonator rethermalization. For resonant coupling, the quantized Hamiltonian in the Jaynes–Cummings form is[42]

$$H = \omega_m \left(a^\dagger a + S_+ S_-\right) + \lambda(S_+ + S_-)\left(a^\dagger + a\right) \qquad (3)$$

where $a^\dagger$ ($a$) is the creation (annihilation) operator for the mechanical mode and $S_\pm$ are the ladder operators for the spin state. The spin relaxation term in equation (2) takes the form $\mathcal{L}_\Gamma \rho = (2T_1)^{-1} \mathcal{D}[S_-]\rho + (2T_2^*)^{-1}(S_z \rho S_z - \rho)$, where $\mathcal{D}[S_-]\rho = (2S_- \rho S_+ - S_+ S_- \rho - \rho S_+ S_-)$ is the Lindblad superoperator and $T_1$ ($T_2^*$) is the transverse (longitudinal) spin coherence time. The resonator rethermalization is described by $\mathcal{L}_\gamma \rho = \frac{\gamma}{2}(n_{th} + 1)\mathcal{D}[a]\rho + \frac{\gamma}{2} n_{th} \mathcal{D}[a^\dagger]\rho$.

Within this two-state model, an analytic expression for the steady state phonon number $n_f$ can be found by using the matrix of second order moments (Supplementary Note 3)[43].

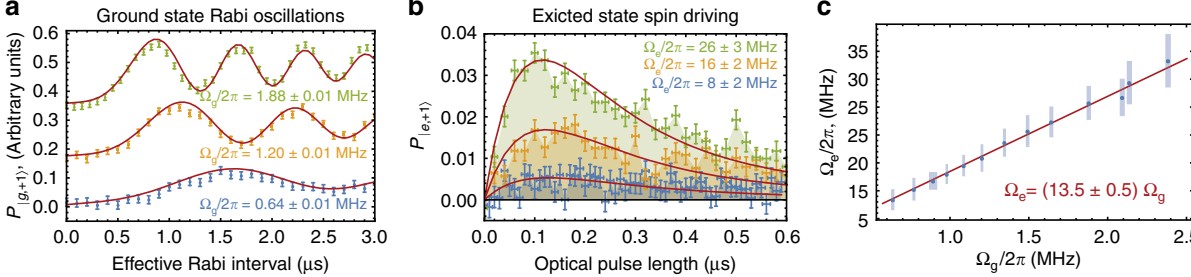

**Figure 3 | Quantifying the excited state spin–strain coupling.** (**a**) Mechanically driven Rabi oscillations within the ground state (GS) that have been fit using the procedure described in Supplementary Note 2. (**b**) Population in $|e, +1\rangle$ plotted as a function of $\tau_{opt}$. The red curves are least squares fits to a seven-level master equation model of the measurement. The data in **a,b** were measured on a single device with an NV centre ensemble, and error bars are from the s.d. in photon counting. (**c**) The excited state mechanical driving field plotted against the GS mechanical driving field and fit with a linear scaling. Each point corresponds to a single measurement, and error bars are standard error from the fits.

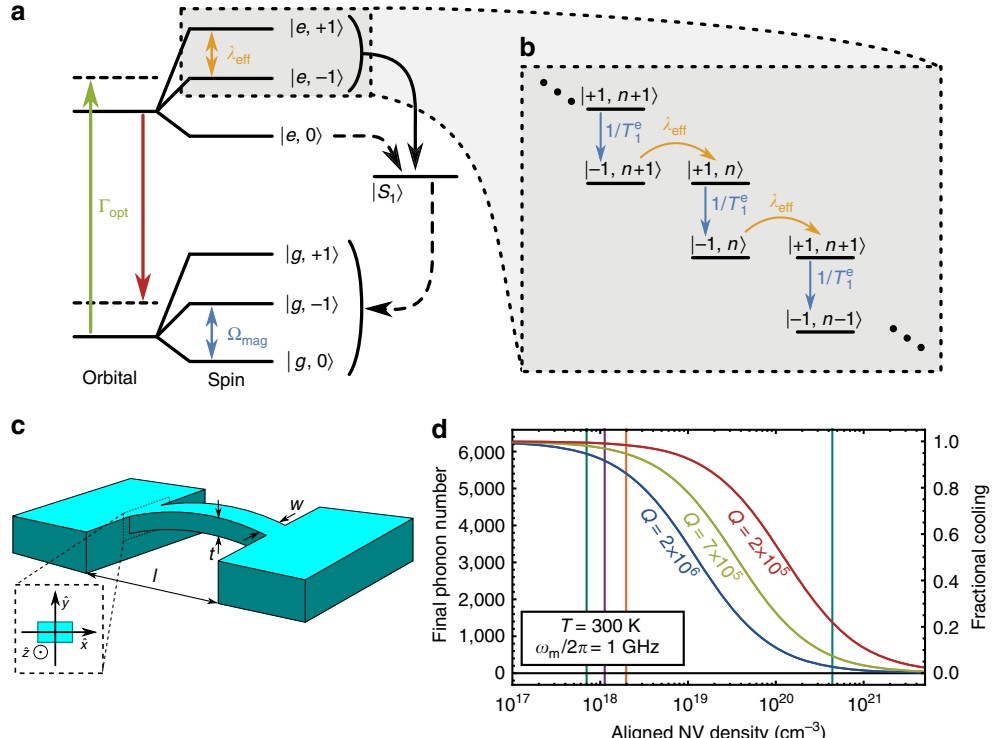

**Figure 4 | A dissipative protocol for cooling a mechanical resonator.** (**a**) The seven NV centre orbital and spin states at room temperature. Fast (slow) transitions are indicated by solid (dashed) one-way arrows. Coherent couplings are indicated by two-way arrows. (**b**) The toy model depiction of the proposed cooling protocol. (**c**) Schematic of a doubly-clamped beam. (**d**) Final phonon number achieved by the cooling protocol as a function of the density of properly aligned NV centers. Vertical lines indicate densities that have been realized in single-crystal diamonds ($7.0 \times 10^{17} \, \text{cm}^{-3}$ (ref. 31), $1.1 \times 10^{18} \, \text{cm}^{-3}$ (ref. 58), $2.0 \times 10^{18} \, \text{cm}^{-3}$ (ref. 56)) and nanodiamonds ($4 \times 10^{20} \, \text{cm}^{-3}$ (ref. 57)).

Under the secular approximation and working in the limit $\gamma, \lambda \gg 1/T_1, 1/T_2^*$, the dynamical equation for the phonon occupancy $n = \langle a^\dagger a \rangle$ can be simplified to

$$\frac{dn}{dt} = \gamma(n_{\text{th}} - n) - \frac{4\lambda^2}{2/T_2^* + 1/T_1} n. \qquad (4)$$

For an ensemble of $N$ spins coupled to the resonator but not to one another, each spin will add an additional damping term to the resonator dynamical equation. This allows us to rewrite the last term in equation (4) as $\sum_{i=1}^{N} \frac{4\lambda_i^2}{2/T_{2,i}^* + 1/T_{1,i}} n$. If each spin within the ensemble has the same $T_1$ and $T_2^*$, we can factorize this expression and replace the individual $\lambda_i$ with an effective ensemble-resonator coupling $\lambda_{\text{eff}} = \sqrt{\sum_{i=1}^{N} \lambda_i^2}$. For the case of uniform coupling, this simplifies to $\lambda_{\text{eff}} = \sqrt{N}\lambda$, which is equivalent to the effective coupling in the Tavis–Cummings model[44,45]. Solving for the steady state of the system then gives

$$n_{\text{f}} = \frac{\gamma n_{\text{th}}}{\gamma + \frac{4\lambda_{\text{eff}}^2}{2/T_2^* + 1/T_1}}. \qquad (5)$$

The problem now becomes mapping the seven-level NV centre structure pictured in Fig. 4a onto this two-state spin system. We do this by distilling the seven-level landscape to a toy model that contains only the two-states that couple to the mechanical resonator, $|e, +1\rangle$ and $|e, -1\rangle$, as shown in Fig. 4b. Within this simplified landscape, we assign $T_2^*$ to be the ES coherence time ($T_{2e}^* = 6.0 \, \text{ns}$ (ref. 46)) and $T_1$ to be the ES lifetime of $|e, +1\rangle$ ($T_{1e} = 6.89 \, \text{ns}$ (ref. 47)). At any one moment, only a fraction $\alpha$ of the spins within the ensemble will be in the proper $\{|e, +1\rangle, |e, -1\rangle\}$ subspace to participate in the cooling.

We account for this by modifying $\lambda_{\text{eff}} \rightarrow \sqrt{\alpha}\lambda_{\text{eff}}$. To determine $\alpha$, we solve for the $7 \times 7$ density matrix describing the steady state of the ensemble in the absence of the mechanical resonator, calculate the population difference between $|e, -1\rangle$ and $|e, +1\rangle$, and obtain $\alpha = 0.017$ for optimized control fields (Supplementary Note 4).

Elasticity theory provides a means of calculating the remaining device parameters. For a doubly-clamped beam of length $l$, thickness $t$ and width $w$, we compute $\lambda_{\text{eff}}$ from the strain because of the zero-point motion of the resonator $\epsilon_0(y, z)$ with coordinates as defined in Fig. 4c (see Methods). For a uniform distribution of properly aligned NV centres at a density $v$, we obtain[41,48]

$$\lambda_{\text{eff}} = d_\perp^e \sqrt{\alpha v w} \sqrt{\int_0^l \int_{-t/2}^{t/2} \epsilon_0^2(y, z) \, dy \, dz}. \qquad (6)$$

Evaluating equation (6), we find that $\lambda_{\text{eff}}$ is independent of $w$ and scales as $\lambda_{\text{eff}} = G_0 \sqrt{vt}/l$ where $G_0 = d_\perp^e \sqrt{\hbar \alpha \kappa_0 / E}$, $\kappa_0 = 120 \, \text{GHz} \cdot \mu\text{m}$, and $E = 1{,}200 \, \text{GPa}$ is the Young's modulus of diamond. The frequency of the resonator's fundamental mode scales as $\omega_{\text{m}} = \kappa_0 t / l^2$. As described in Supplementary Note 5, higher order mechanical modes are spectrally isolated from the NV centre spin dynamics in the devices considered here[17,41]. For any thin-beam resonator in the resolved-sideband regime ($\omega_{\text{m}}/2\pi > 1/T_{2e}^* \sim 170 \, \text{MHz}$), the fractional cooling $n_{\text{f}}/n_{\text{th}}$ is insensitive to the physical dimensions of the resonator because the size of the ensemble scales with the size of the resonator. This can be seen by rewriting equation (5) as $n_{\text{f}}/n_{\text{th}} = (1 + \chi)^{-1}$, where $\chi = \frac{4\hbar Q v \alpha (d_\perp^e)^2}{E(2/T_{2e}^* + 1/T_{1e})}$ is independent of the resonator dimensions. For illustrative purposes, we choose to examine a resonator with a

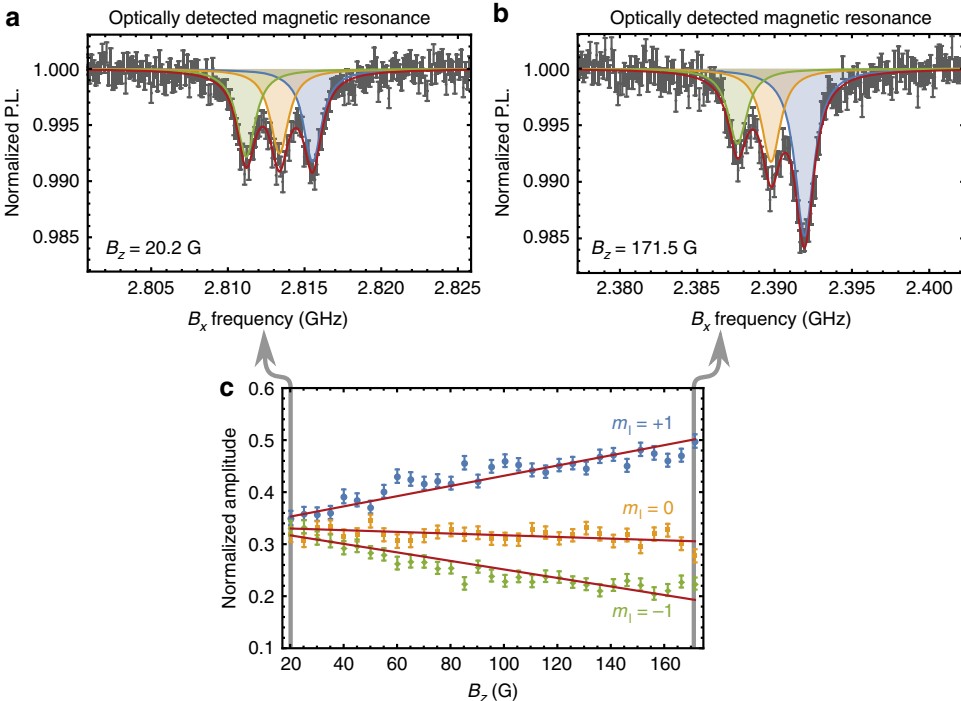

**Figure 5 | Calibrating $B_z$-dependent nuclear polarization.** (**a,b**) Normalized photoluminescence (P.L.) plotted as a function of the magnetic driving field carrier frequency for (**a**) $B_z = 20.2$ G and (**b**) $B_z = 171.5$ G. The solid line in each plot is a least squares fit to the sum of three Lorentzians. The data in **a,b** was measured on a single device with an NV centre ensemble, and error bars are from the s.d. in photon counting. (**c**) Normalized amplitude of each $m_l$ hyperfine sublevel as a function of $B_z$. The solid lines are least squares fits to a linear model, each point corresponds to a single measurement, and error bars are standard error from the fits.

$\omega_{\mathrm{m}}/2\pi = 1$ GHz fundamental mode and assume fully polarized nuclear spins[49]. Potential device dimensions then become $(l, t) = (1.9, 0.19)$ μm. Finally, phonon–phonon interactions limit the $Q$ of an ideal diamond mechanical resonator at room temperature. For modes satisfying $\omega_{\mathrm{m}}/2\pi > 1/T_{2e}^*$, the maximum $Q = 2 \times 10^6$ is independent of $\omega_{\mathrm{m}}$ (ref. 50), and we now have all the parameters needed to study the performance of the protocol.

At this point, we note that distilling a seven-state model to the toy model we employ certainly requires validation. To justify our simplified model, we calculate the cooling predicted within the toy model and compare this both to an analytical Lamb–Dicke treatment of the seven-level model[14,51] and to numerical simulations of a small number of seven-level NV centres coupled to a resonator. Because of the exponential growth of the Hilbert space, full seven-level numerical simulations were performed on the Titan supercomputer at Oak Ridge National Laboratory, with the most intensive simulations taking ∼$10^4$ core-hours. Comparing the toy model and Lamb–Dicke results to the numerical simulations, we determined that the two-level distillation outperforms the Lamb–Dicke approach in all test cases and provides an upper bound on $n_{\mathrm{f}}$ (Supplementary Note 6). This indicates that the proposed protocol cools a resonator more efficiently than our toy model predicts[52–55].

**Cooling performance.** The lowest phonon occupancy that can be reached depends strongly on the density of properly aligned NV centres $v$. For instance, Choi, *et al.* reported measurements of an NV centre ensemble with $v = 2.0 \times 10^{18}$ cm$^{-3}$ in single-crystal diamond[56]. For this density and $Q = 2 \times 10^6$, we find that the proposed protocol cools a room temperature resonator to $n_{\mathrm{f}} = 0.86 n_{\mathrm{th}}$. Using the same $Q$ and the density of $v = 4 \times 10^{20}$ cm$^{-3}$ reported by Baranov, *et al.* in nanodiamonds[57], however, the protocol can cool to $n_{\mathrm{f}} = 0.03 n_{\mathrm{th}}$.

Increasing the size of the ensemble can thus dramatically increase the protocol's cooling power. The magnetic field noise from paramagnetic impurities will also grow with $v$, degrading the GS coherence time. However, for large magnetic driving fields, this cooling protocol does not require a lengthy GS coherence time (Supplementary Note 4). The only coherence time that effects the protocol is $T_{2e}^*$, which is not expected to change with the defect density[34,37]. This means that large NV centre densities could in principle be used to cool a resonator with the ES spin–strain interaction. To study how increasing $v$ affects the protocol, we plot $n_{\mathrm{f}}$ against $v$ in Fig. 4d for several different $Q$-values. For reference, we have included lines marking values of $v$ that have been realized in single-crystal diamonds[31,56,58] and in nanodiamonds[57]. The limiting density of NV centres in a single-crystal diamond nanostructure is currently unknown. Furthermore, while high defect densities have been shown to degrade the $Q$ of $\omega_{\mathrm{m}}/2\pi \sim 10$ kHz frequency resonators[59], it remains to be seen how the gigahertz frequency resonators of interest here will be affected by the incorporation of a dense defect ensemble. These questions motivate future experimental work.

## Discussion

The insensitivity of the proposed protocol to the GS coherence time makes an ES cooling protocol an attractive and practical route to cooling a room temperature mechanical resonator with NV centres. Alternative approaches that use the GS spin–strain coupling[18,19,25] are incompatible with the collective enhancement from a dense ensemble that makes the proposed protocol viable. Although the GS inhomogeneous dephasing time $T_{2g}^*$ can be ∼μs long in high purity diamonds, $T_{2g}^*$ scales roughly as ∼$1/v$ in bulk diamond and can be <100 ns for dense ensembles[30,31]. Within a nanostructure, effects such as exchange narrowing and the

truncation of the spin bath mitigate this reduction in $T_{2g}^*$ (refs 57,60) and make it difficult to predict the decrease in $T_{2g}^{*g}$ inside a doubly-clamped beam. Nevertheless, we can roughly compare the ES and GS spin–strain interactions by calculating $\eta$ using coherence times measured in bulk diamond (Supplementary Note 7). For a moderate NV centre density of $v = 2.8 \times 10^{13}\,\mathrm{cm}^{-3}$ (ref. 18), the single-spin cooperativity for the ES spin–strain interaction is 2.4-times larger than in the GS, and for $v = 7.0 \times 10^{17}\,\mathrm{cm}^{-3}$ (ref. 31), $\eta$ is 19-times larger in the ES. In both cases, the ES offers the more efficient route to cooling, and as the collective enhancement grows, the ES interaction becomes increasingly more efficient than the GS interaction. A dense ensemble coupled via the ES spin–strain interaction thus becomes the more promising route to cooling a room temperature mechanical resonator with NV centres.

It is important to note that this analysis of the proposed protocol only applies for operation at room temperature. Reducing the bath temperature will lower $n_{th}$ and would thus ideally lower $n_f$. However, the ES coherence time is limited by the ES motional narrowing rate, which increases as the bath temperature decreases[34,37]. This is expected to lead to a reduction in $T_{2e}^*$ at lower temperatures, followed by a complete loss of ES spin coherence below $\sim 150\,\mathrm{K}$ (ref. 32). As seen from equation (5), a reduction in $T_{2e}^*$ will lead to a loss of cooling power. For cryogenic starting temperatures, it thus becomes necessary to use either the GS spin-phonon interaction or the orbital-strain interaction to cool a mechanical resonator with NV centres.

Finally, $n_f$ could be lowered further by simultaneously implementing an optomechanical cooling protocol[8–12] alongside the proposed protocol. Optomechanical cooling has been demonstrated to cool gigahertz frequency resonators to $n_f \sim 0.01 n_{th}$ (refs 6,61). The cooling rate from an optimized realization of the proposed protocol would combine additively with the optomechanical cooling rate, allowing the two complementary techniques to operate in conjunction and enhance the total cooling.

In conclusion, we have proposed a dissipative protocol for cooling a room temperature mechanical resonator that utilizes an ensemble of NV centre spins to realize a collective enhancement in the spin-phonon coupling. After experimentally determining that the spin–strain coupling in the room temperature ES is $13.5 \pm 0.5$ times stronger than the GS spin–strain coupling, we analysed the performance of the cooling protocol. For very dense NV centre ensembles, the proposed protocol can cool a room temperature resonator to a fraction of its thermal phonon occupancy. These results shed further light on the orbitally-averaged room temperature ES of the NV centre and demonstrate a practical path towards cooling a room temperature mechanical resonator with NV centres.

## Methods

**Sample details.** Our HBAR consists of a 2.5 µm-thick ZnO piezoelectric film sandwiched between a 25/200 nm Ti/Pt ground plane and a 250 nm thick apodized Al top electrode, all sputtered on one face of a ⟨100⟩ single-crystal diamond substrate. Applying a high-frequency voltage to this transducer launches acoustic waves into the diamond, which then serves as a Fabry–Pérot cavity to generate a comb of standing wave resonances. By apodizing the shape of the Al electrode, we mitigate the loss of power into lateral modes formed across the diameter of the HBAR. The antenna fabricated on the opposite diamond face was patterned from 25/225 nm Ti/Pt with a lift-off process.

The CVD-grown diamond used in these measurements contained NV centres at a density of $\sim 4 \times 10^{14}\,\mathrm{cm}^{-3}$ as purchased. Our measurements thus address an ensemble of $\sim 70$ NV centres oriented with their symmetry axis parallel to $B_z$. NV centres of different orientations are spectrally isolated and contribute only a constant background to the measurements.

**Spectrum fitting.** The spectrum pictured in Fig. 2 was fit to the function

$$
\begin{aligned}
&P_{|+1\rangle} \\
&= c_e \left( \frac{a_+[B_z]}{\frac{1}{4}\Gamma_e^2 + \left(B_z - B_0 + A_{||}^e/\gamma_{NV}\right)^2} + \frac{a_0[B_z]}{\frac{1}{4}\Gamma_e^2 + (B_z - B_0)^2} + \frac{a_-[B_z]}{\frac{1}{4}\Gamma_e^2 + \left(B_z - B_0 - A_{||}^e/\gamma_{NV}\right)^2} \right) \\
&+ c_g \left( \frac{a_+[B_z]}{\frac{1}{4}\Gamma_g^2 + \left(B_z - B_0 + A_{||}^g/\gamma_{NV}\right)^2} + \frac{a_0[B_z]}{\frac{1}{4}\Gamma_g^2 + (B_z - B_0)^2} + \frac{a_-[B_z]}{\frac{1}{4}\Gamma_g^2 + \left(B_z - B_0 - A_{||}^g/\gamma_{NV}\right)^2} \right)
\end{aligned}
\tag{7}
$$

where $c_e$ and $c_g$ are constant amplitudes that quantify the driven spin contrast for the ES and GS resonances, $a_i[B_z]$ are field-dependent scaling factors that account for the dynamic nuclear polarization of the hyperfine sublevels[49], $\Gamma_e$ ($\Gamma_g$) is the

**Table 1 | Relaxation rates used in our seven-level master equation model[47].**

| Parameter | Value (MHz) | Relaxation from |
|---|---|---|
| $k_{42}$ | $65.3 \pm 1.6$ | ES $|\pm 1\rangle$ to GS $|\pm 1\rangle$ |
| $k_{31}$ | $64.9 \pm 1.5$ | ES $|0\rangle$ to GS $|0\rangle$ |
| $k_{45}$ | $79.8 \pm 1.6$ | ES $|\pm 1\rangle$ to $|S_1\rangle$ |
| $k_{35}$ | $10.6 \pm 1.5$ | ES $|0\rangle$ to $|S_1\rangle$ |
| $k_{52}$ | $2.61 \pm 0.06$ | $|S_1\rangle$ to GS $|\pm 1\rangle$ |
| $k_{51}$ | $3.00 \pm 0.06$ | $|S_1\rangle$ to GS $|0\rangle$ |

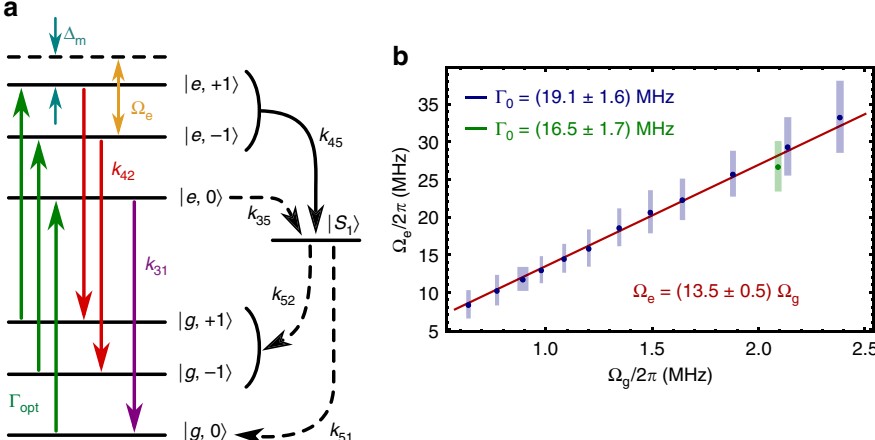

**Figure 6 | Transitions and rates used in the seven-level model.** (**a**) States and transitions included in our seven-level master equation model. The $k_{ij}$ rates are listed in Table 1. (**b**) Excited state mechanical driving field $\Omega_e$ plotted as a function of the ground state mechanical driving field $\Omega_g$ with the data labelled by the optical pumping rate $\Gamma_0$ during each measurement. Each point corresponds to a single measurement, and error bars are standard error from the fits.

FWHM of the ES (GS) resonances, $B_0$ is the resonant field for the $m_I = 0$ hyperfine sublevel, and the other parameters are as defined above. Of these variables, $c_i$, $\Gamma_i$, and $B_0$ are free parameters in our fitting procedure.

We calibrate $a_i[B_z]$ by performing hyperfine-resolved magnetically-driven electron spin resonance (ESR) measurements within the NV centre GS at different values of the magnetic bias field $B_z$. This is done by fixing $B_z$ and monitoring the NV centre photoluminescence as the carrier frequency of a magnetic driving field oriented along $B_x$ is swept through the $|g, 0\rangle \leftrightarrow |g, -1\rangle$ spin resonance. We fit the resulting curves to the function

$$P = C\left(\frac{A_+}{\frac{1}{4}\Gamma_g^2 + \left(B_z - B_0 + A_\parallel^g\right)^2} + \frac{A_0}{\frac{1}{4}\Gamma_g^2 + \left(B_z - B_0\right)^2} + \frac{A_-}{\frac{1}{4}\Gamma_g^2 + \left(B_z - B_0 - A_\parallel^g\right)^2}\right) + P_0$$

(8)

where $P$ is the measured photoluminescence, $C$ accounts for the driven spin contrast, $A_i$ is the relative amplitude of each hyperfine sublevel, $P_0$ is the background photoluminescence, and we fix $\sum A_i = 1$. Figure 5a,b shows ESR curves measured at $B_z = 20.2$ G and 171 G. We have used the values of $P_0$ returned from the fits to normalize the photoluminescence.

Figure 5c shows the normalized amplitude of each nuclear sublevel plotted as a function of $B_z$. As expected, the nuclear polarization increases in the direction of the ES level anti-crossing at $B_z^{LAC} = 507$ G. In this figure, we have fit each curve to a straight line with a fixed $y$-intercept of $\frac{1}{3}$ to obtain the linear scaling functions $a_i[B_z]$ in equation (7). The sum of these scaling functions satisfies $\sum a_i[B_z] = 1$.

**Seven-level master equation model.** The master equation used to model our ES spin driving measurements is derived in the room temperature NV centre basis defined by the states $\{g_{+1}, g_0, g_{-1}, e_{+1}, e_0, e_{-1}, S_1\}$ where, within the $e$ and $g$ subspaces, a subscript denotes the $m_s$ value. The $7 \times 7$ density matrix evolves according to ($\hbar = 1$)

$$\dot{\rho} = -i[H, \rho] + \mathcal{L}_\Gamma \rho.$$

(9)

In the rotating frame, the Hamiltonian is given by

$$H = \frac{\Omega_e}{2}\left(|e_{+1}\rangle\langle e_{-1}| + |e_{-1}\rangle\langle e_{+1}|\right) + \Delta_m |e_{+1}\rangle\langle e_{+1}|$$

(10)

where $\Delta_m$ is the mechanical detuning. The incoherent NV centre processes are described by the superoperator

$$\mathcal{L}_\Gamma \rho = \Gamma_{opt} \sum_{i=\pm 1, 0} L_{g_i, e_i} + k_{42} \sum_{i=\pm 1} L_{e_i, g_i} + k_{45} \sum_{i=\pm 1} L_{e_i, S_1} + k_{52} \sum_{i=\pm 1} L_{S_1, g_i}$$
$$+ k_{31} L_{e_0, g_0} + k_{35} L_{e_0, S_1} + k_{51} L_{S_1, g_0} + \frac{1}{T_{2e}^*} \sum_{i=\pm 1, 0} L_{e_i, e_i}$$

(11)

where we define

$$L_{if} \rho = |f\rangle\langle i|\rho|i\rangle\langle f| - \frac{1}{2}\left(|i\rangle\langle i|\rho + \rho|i\rangle\langle i|\right).$$

(12)

Here, $\Gamma_{opt}$ is the optical pumping rate of our 532 nm laser, $T_{2e}^* = 6.0 \pm 0.8$ ns is the ES coherence time[46], and the relaxation rates $k_{ij}$ are listed in Table 1. Figure 6a summarizes this landscape[47].

Because optical initialization does not generate a pure state, we first simulate the optical pumping process to obtain an initialized density matrix. To do so, we start with a thermal state $\rho_{NV} = \frac{1}{3}\left(\sum_{i=\pm 1, 0} |g_i\rangle\langle g_i|\right)$ and apply equation (9) with $\Omega_e$, $\Delta_m = 0$ and $\Gamma_{opt} \neq 0$ for 10 μs. We take the resulting density matrix and apply equation (9) for 5 μs with $\Omega_e$, $\Delta_m$, $\Gamma_{opt} = 0$ to simulate the relaxation to $|g\rangle$. A simulated π-pulse then swaps $\rho_{22}$ and $\rho_{33}$, providing the appropriate starting density matrix $\rho_0$ for a given $\Gamma_{opt}$. From $\rho_0$, we also extract the minimum and maximum spin contrast ($s_{min} = \langle g_0|\rho_0|g_0\rangle$ and $s_{max} = \langle g_{-1}|\rho_0|g_{-1}\rangle$), which allow us to properly normalize our simulations.

Next, we model the measurement of $S_2$, the spin re-initialization. To do so, we apply equation (9) to $\rho_0$ with $\Omega_e$, $\Delta_m = 0$ and $\Gamma_{opt} \neq 0$ for a length of time $\tau_{opt}$. Allowing the spin to relax as before gives us the measured density matrix $\rho_2$. We normalize $\langle g_{-1}|\rho_2|g_{-1}\rangle$ using $s_{min}$ and $s_{max}$, and repeat this simulation as a function of $\tau_{opt}$ to obtain a simulated $S_2$ curve.

To account for spatial inhomogeneities in the optical power within the NV centre ensemble, we perform a weighted average of this simulation over the point spread function (PSF) of our microscope. We approximate the PSF by the function

$$\Gamma_{opt}(z) = \Gamma_0 \left\{\frac{\sin\{\kappa[z_0](z - z_0)\}}{\kappa[z_0](z - z_0)}\right\}^2$$

(13)

where $\Gamma_0$ is the peak optical pumping rate, $\kappa[z_0]$ defines the depth-dependent PSF width[15], $z$ is the distance below the diamond surface, and $z_0 = 7.9 \pm 0.9$ μm is the focus depth of the PSF. An ensemble measurement is then given by

$$S_2^{ens}(\tau_{opt}) = \frac{\int_0^\infty S_2[\tau_{opt}, \Gamma_{opt}(z)]\,dz}{\int_0^\infty \Gamma_{opt}(z)\,dz}.$$

(14)

We discretize this integral to make it computationally tractable and perform a least squares fit of $S_2^{ens}(\tau_{opt})$ to the measured data. $\Gamma_0$ is the only free parameter in the fitting procedure.

With the exception of the datum indicated in Fig. 6b, all of the measurements were taken at the same optical power. We simultaneously fit each of these $S_2$ curves and find $\Gamma_0 = 19.1 \pm 1.6$ MHz. For the measurement at a different optical power, we find $\Gamma_0 = 16.5 \pm 1.7$ MHz.

To extract $\Omega_e$, we then fix $\Gamma_0$ and repeat this procedure with $\Omega_e \neq 0$ to simulate the $S_2 - S_1$ measurement pictured in Fig. 3b. To account for inhomogeneities in the mechanical driving field, we must also include the spatial profile of the strain standing wave inside the weighted average. This function takes the form $\Omega_e(z) = \Omega_0 |\sin[2\pi z/\lambda]|$ where $\lambda = 31 \pm 4$ μm is the wavelength of the strain wave. Defining the results of such a simulation as $P_{|+1\rangle}(\Omega_e, \Delta_m)$, we account for the hyperfine sublevels by computing the sum

$$P_{|+1\rangle}(\Omega_e) = a_{+1}P_{|+1\rangle}(\Omega_e, 0) + a_0 P_{|+1\rangle}\left(\Omega_e, A_\parallel^e\right) + a_{-1}P_{|+1\rangle}\left(\Omega_e, 2A_\parallel^e\right)$$

(15)

where the normalized amplitudes ($\sum a_i = 1$) account for nuclear spin polarization and have been measured separately via magnetic ESR. A least squares fit of equation (15) to the data then provides $\Omega_e$. Here, $\Omega_e$ is the only free parameter in the fitting procedure.

When fitting the relation between $\Omega_e$ and $\Omega_g$ (Fig. 3c), we fix the $y$-intercept of the linear fitting function to be zero.

**Elasticity theory.** To analyse how strain couples to NV centres within a resonator, we start by assuming that the NV centres are aligned with the direction of beam deflection such that the strain in an oscillating beam is entirely perpendicular to the NV centre symmetry axis. We then use elasticity theory to derive the scaling laws quoted above[17,41].

The wave equation for doubly-clamped beams is

$$\rho_d A \frac{\partial^2}{\partial t^2}\phi(t, z) = -EI \frac{\partial^4}{\partial z^4}\phi(t, z)$$

(16)

where $\phi(t, z)$ is the transverse displacement in the $y$-direction, $\hat{z}$ points along the beam as indicated in Fig. 4c, $A = wt$ is the cross-sectional area of the resonator, $E = 1,200$ GPa is the Young's modulus of diamond, $\rho_d = 3.515$ g cm$^{-3}$ is the mass density of diamond, and $I = wt^3/12$ is the resonator's moment of inertia. Solutions are of the form $\phi(z, t) = u(z)e^{-i\omega t}$ where

$$u_n(z) = a_n(\cos k_n z - \cosh k_n z) - b_n(\sin k_n z - \sinh k_n z),$$

(17)

and the allowed $k$-vectors satisfy $\cos(k_n z)\cosh(k_n z) = 1$. The wave vector and amplitudes of the fundamental mode satisfy $k_0 l = 4.73$ and $a_0/b_0 = 1.0178$.

We normalize $u_n(z)$ by setting the free energy of the beam equal to the zero point energy of the mode:

$$W = \frac{1}{2}EI \int_0^L \left(\frac{\partial^2 u_n}{\partial z^2}\right)^2 dz = \frac{1}{2}\hbar\omega_n$$

(18)

where the eigenfrequencies of the resonator are given by $\omega_n = k_n^2\sqrt{EI/\rho_d A}$. This expression for $\omega_n$ can be simplified to $\omega_n = \kappa_n t/l^2$, where $\kappa_n = (k_n l)^2\sqrt{E/12\rho_d}$. For the fundamental mode, $\kappa_0 = 120$ GHz · μm as quoted above.

The spin-phonon coupling for a single NV centre located at $(y, z)$ is given by $\lambda = d_\perp \epsilon_0(y, z)$ where $\epsilon_0(y, z) = -y\frac{\partial^2}{\partial z^2}u_n(z)$ is the strain from the zero point motion of the resonator mode. Here, the $y$-axis is zeroed at the neutral axis of the resonator. To compute the effective ensemble-resonator coupling, we assume a uniform distribution of properly aligned NV centres within the resonator and sum the individual couplings in quadrature according to $\lambda_{eff} = \sqrt{\alpha}\sqrt{\sum_{i=1}^N \lambda_i^2}$, which gives equation (6).

**Data availability.** The data that support the findings of this study are available from the corresponding author on reasonable request.

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

## Acknowledgements

Research support was provided by the Office of Naval Research (ONR) (Grant N000141410812). Device fabrication was performed in part at the Cornell NanoScale Science and Technology Facility, a member of the National Nanotechnology Coordinated Infrastructure, which is supported by the National Science Foundation (Grant ECCS-15420819), and at the Cornell Center for Materials Research Shared Facilities which are supported through the NSF MRSEC program (DMR-1120296). Numerical simulations were performed in part at the Centre for Nanoscale Materials, a U.S. Department of Energy Office of Science User Facility under contract no. DE-AC02-06CH11357. This research used resources of the Oak Ridge Leadership Computing Facility at the Oak Ridge National Laboratory, which is supported by the Office of Science of the U.S. Department of Energy under Contract No. DE-AC05-00OR22725.

## Author contributions

E.R.M. and G.D.F. developed the concept and procedure for the experiment and the proposed cooling protocol. E.R.M. performed the experiments and analysed the data. E.R.M., M.O., S.K.G. and G.D.F. developed the toy model treatment of the proposed cooling protocol. M.O. and S.K.G. developed and performed the numerical simulations that validate the toy model treatment of the cooling protocol. E.R.M., M.O, S.K.G. and G.D.F. prepared the manuscript.

## Additional information

**Competing financial interests:** The authors declare no competing financial interests.

DOI: 10.1038/ncomms16166      **OPEN**

# Erratum: Cooling a mechanical resonator with nitrogen-vacancy centres using a room temperature excited state spin–strain interaction

E.R. MacQuarrie, M. Otten, S.K. Gray & G.D. Fuchs

Nature Communications 8:14358 doi: 10.1038/ncomms14358 (2017); Published 6 Feb 2017; Updated 29 Nov 2017.

This Article contains a typographical error in the sentence preceding Eq. 4. This sentence should read, 'Under the secular approximation and working in the limit $\gamma, \lambda \ll 1/T_1, 1/T_2^*$, the dynamical equation for the phonon occupancy $n = \langle a^\dagger a \rangle$ can be simplified to'; that is, with a much-less-than symbol rather than a much-greater-than symbol.

There is a further error in Fig 4. The label on the upper state of the right-most transition in Fig. 4b should be "$|+1, n-1\rangle$" not "$|+1, n+1\rangle$".

