## [Peer Review File · Nature Communications]

Reviewers' comments:

Reviewer #1 (Remarks to the Author):

The paper consists of two parts. The first part reports on an experiment for measuring the spin-strain coupling constant in the excited state of a NV center in diamond. This coupling constant was unknown so far and it is found to be more than 10 times larger than in the ground state. In the second part the authors propose and analyze a schemes for using this large strain coupling to cool the vibrations of a mechanical resonator by an optically pumped dense ensemble of NV centers. It is estimated that by using this scheme, a cooling factor of about ~ 40 (starting from room-temperature) could be achieved.

The experimental observation of the very large excited-state coupling is certainly very interesting. As the authors point out correctly, this could be used to boost the weak strain coupling in the ground state, while not being severely affected by the problem of inhomogeneous broadening of orbital states. The authors address an intriguing question, namely whether one can combine these two features to enhance the insignificant cooling power of a single spin, by coupling the mechanical resonator to a large ensemble of NV centers. However, the presented analysis is not very transparent and the assumed parameters for evaluating the overall cooling are not convincing. In particular, it seems that even a very moderate cooling factor of about 10 requires quite extreme densities, at which even the model of individual NV centers becomes questionable. Any reduction of the here assumed optimistic parameters in a real experiment would make the cooling rate insignificant.

In summary, while I find the experimental results as well as the general idea of an ensemble-enhanced cooling scheme very interesting, it seems that in the end the proposed approach is not very useful in practice. In addition, there are several problematic assumptions in the derivation of the cooling rate and therefore I cannot recommend the paper for publication in its present form. A more detailed list of comments is given below.

-) First sentence in the introduction should be changed. Active cooling of a mechanical resonator does usually not improve the measurement sensitivity, since the reduced fluctuations are compensated by a larger damping rate. In the cited references the resonator is still in equilibrium with the support temperature.

-) For the analysis of the cooling dynamics the authors approximate each 7-level NV center by an effective two level system, and describe the collective coupling to the mechanical mode by a collective spins master equation. In my opinion this approach is both wrong and also unnecessary. For example, since the NV centers decay and dephase independently of each other, a collective spin description for the decay or dephasing processes is invalid. However, since even the collective coupling λ_{eff} is smaller than the decay rate, there are no collective Rabi-oscillations and the system can safely be described by independent NV centers. This means that it is enough to evaluate the cooling rate for each center and add up all the rates with the appropriate average over the coupling constants. There are many papers in the literature that evaluate mechanical cooling rates using two or multilevel system, which can be adapted for the current problem without the need for a supercomputer. Such an approach would certainly be more appropriate and easier to follow.

-) The cooling rate is reduced by the very low value of $\alpha=0.017$. It seems to me that it should be possible to improved this parameter by a factor of 10 or so by optimizing the driving. Is this true?

-) The authors use a thin beam theory to evaluate the scalings of the resonator properties like the frequency and coupling constant. However, the resonator dimension assumed for the evaluation of

the cooling rate are no longer thin beams. Since the predicted cooling is very borderline, either a detailed numerical study of the vibrational mode, or 'thin-beam-like' dimensions should be used for estimating the parameters.

-) In the outlook the authors state that their cooling scheme could be supplemented by other optical cooling schemes to reach the ground state. This is not true (or at least misleading) since the cooling factors will not multiply each other, i.e. if the optical scheme already cools by a factor of 1000 the additional cooling rate provided by the NV centers is negligible.

Reviewer #2 (Remarks to the Author):

Manuscript entitled "Cooling a Mechanical Resonator with a Nitrogen-Vacancy Center Ensemble Using a Room Temperature Excited State Spin-Strain Interaction" by MacQuarrie et al. reports an experimental study on the excited state (ES) spin-strain coupling strength and a theoretical proposal and analysis on the use of an ensemble of NV centers to cool a mechanical resonator through the ES spin strain coupling.

The experimental study seems to be solid and provides valuable information on the ES spin-strain coupling. The theoretical proposal is also quite interesting. The subject matter of cooling a mechanical resonator via coupling to an ensemble of defect centers should be of significant interest to both the opto-mechanics and the spin-mechanics communities. My questions and comments here are mostly on the comparison of the proposed ensemble cooling via ES spin-strain coupling with ensemble cooling via ground state (GS) spin-strain coupling and also with ensemble cooling via a phonon-assisted optical transition. These comparisons should further strengthen the paper, making it suitable for publication in Nature Communications.

1) An advantages of using the ES spin-strain coupling instead of GS spin-strain coupling for cooling is the stronger ES coupling strength (about 13.5 times stronger, as shown by the authors). However, I think the figure-of-merit in this case is the cooperativity for a single spin. Although as the authors pointed out, at relatively large densities, the spin decoherence time gets significantly shorter, it is still informative to have a direct comparison of the cooperativity at a moderate NV density. My concern is that T_1 time for the ES spin transition is quite short (limited by the excited state lifetime). Without a direct comparison, it is difficult to assess whether cooling due to the ES coupling is indeed much more efficient.

2) I understand that the large inhomogeneous broadening of NV optical transitions greatly reduces the effectiveness of using an ensemble of NVs and a phonon-assisted optical transition for cooling purposes. Nevertheless, since the relevant coupling strength through the phonon-assisted optical transition is about four or five orders of magnitude greater than that of the ES spin-strain coupling, it is not clear which one is more efficient. It is probably not realistic to use the NV optical transitions at room temperature. But it is still of significant interest to start the cooling process at low temperature (for example, 80 K or 10 K). In this regard, it will also be useful to make a comparison between ensemble cooling via ES spin-strain coupling and that via phonon-assisted optical transitions under realistic experimental conditions.

Reviewer #3 (Remarks to the Author):

The manuscript by MacQuarrie et al report new scheme allowing the cooling of mechanical resonator by NV centers in diamond. Cooling of micro and nanomechanical systems is a rapidly growing field of research. Authors of this manuscript show that new approach based on coupling of mechanical motion to spins of NV centers allows very efficient cooling.

The novelty of the scheme proposed by MacQuarrie et al is new owing to the use of photoexcited

triplet state with an order of magnitude large coupling to mechanical motion of the resonator. The paper consists of proposal and preliminary experiments allowing to quantify the strength of the spin-phonon interaction.

The manuscript is well written and from my point of view will be of interest for broad community of scientists. I recommend it for publication after addressing following point.

Authors demonstrate high density ensemble of NVs is capable to cool mechanical mode of the resonator. At the same time populating excited state of NV defect will lead to heating of diamond owing to Stokes shifted emission into phonon sideband. Of course this will lead to a heating of the whole diamond and particular mode of the resonator can be decoupled, there is a limit of temperature of operations of the device owing to decreased lifetime of excited state of NV center at elevated T.

Cooling a Mechanical Resonator with a Nitrogen-Vacancy Center Ensemble Using a Room Temperature Excited State Spin-Strain Interaction:
Response to Reviewers

E. R. MacQuarrie, M. Otten, S. K. Gray, and G. D. Fuchs

Reviewer #1:

The paper consists of two parts. The first part reports on an experiment for measuring the spin-strain coupling constant in the excited state of a NV center in diamond. This coupling constant was unknown so far and it is found to be more than 10 times larger than in the ground state. In the second part the authors propose and analyze a schemes for using this large strain coupling to cool the vibrations of a mechanical resonator by an optically pumped dense ensemble of NV centers. It is estimated that by using this scheme, a cooling factor of about ~40 (starting from room-temperature) could be achieved.

The experimental observation of the very large excited-state coupling is certainly very interesting. As the authors point out correctly, this could be used to boost the weak strain coupling in the ground state, while not being severely affected by the problem of inhomogeneous broadening of orbital states. The authors address an intriguing question, namely whether one can combine these two features to enhance the insignificant cooling power of a single spin, by coupling the mechanical resonator to a large ensemble of NV centers. However, the presented analysis is not very transparent and the assumed parameters for evaluating the overall cooling are not convincing. In particular, it seems that even a very moderate cooling factor of about 10 requires quite extreme densities, at which even the model of individual NV centers becomes questionable. Any reduction of the here assumed optimist parameters in a real experiment would make the cooling rate insignificant.

In summary, while I find the experimental results as well as the general idea of an ensemble-enhanced cooling scheme very interesting, it seems that in the end the proposed approach is not very useful in practice. In addition, there are several problematic assumptions in the derivation of the cooling rate and therefore I cannot recommend the paper for publication in its present form. A more detailed list of comments is given below.

Author Reply:

We thank the reviewer for his/her careful reading of our manuscript and constructive comments. The reviewer recognizes our work as “very interesting” and addressing the “intriguing question” of whether an ensemble coupled to a mechanical resonator can enhance the cooling power of a single spin.

We agree with the reviewer that even moderate cooling with our proposed protocol requires optimistic parameters for both the NV centers and the mechanical resonator. Steady progress in diamond nanoresonator fabrication has brought the proposed resonators within reach, and our results motivate the extension of these efforts into the development of resonators containing dense ensembles of NV centers. The “quite extreme” densities we consider have been experimentally realized in $\sim 10 \mu\text{m}$ nanodiamonds [1], suggesting that the incorporation of such defects into diamond nanostructures is possible. More work is necessary to determine the viability of simultaneously realizing proposed conditions, but our assumptions and conclusions are well-founded based on published results. The proposed protocol offers the firmest foundation for cooling a mechanical resonator from room temperature with NV centers.

At present, the reviewer disagrees with some of the assumptions made in calculating the cooling power of the spin ensemble. We understand the concern. To address it, we have further justified and elaborated on our particular approach in the responses below. We are confident of this approach and optimistic that the revised presentation will satisfy the reviewer comments.

[1] Baranov, *et al*, *Small* **7**, 1533-1537 (2011).

Comment 1:

First sentence in the introduction should be changed. Active cooling of a mechanical resonator does usually not improve the measurement sensitivity, since the reduced fluctuations are compensated by a larger damping rate. In the cited references the resonator is still in equilibrium with the support temperature.

Author Reply:

The reviewer notes that cooling mechanical resonator does not usually improve the measurement sensitivity since the reduced fluctuations are compensated by a larger damping rate. We agree. In our manuscript, however, we do not claim cooling will produce increased sensitivity. Rather, we assert that cooling a resonator can "...enhance sensing by lowering the resonator's thermal noise floor." As described by [1], this is indeed the case for both active and passive cooling. The benefits of passive cooling are demonstrated by the citations included in our manuscript (as noted by the reviewer). For active cooling, this enhancement takes the form of an increased linear dynamic range. Here, the linear dynamic range is defined as the amplitude at which nonlinearities set in divided by the amplitude of thermal noise. Active cooling suppresses this thermal noise, creating a larger linear dynamic range for the sensor.

To address this comment, we have rephrased the first sentence to now read "...enhance sensing by lowering the resonator's thermal noise floor and extending a sensor's linear dynamic range." We have also added a citation of [1] to the manuscript, which was inadvertently omitted from the original manuscript.

[1] M. Poggio, *et al*, *PRL*, **99**,017201 (2007).

Comment 2:

For the analysis of the cooling dynamics the authors approximate each 7-level NV center by an effective two level system, and describe the collective coupling to the mechanical mode by a collective spins master equation. In my opinion this approach is both wrong and also unnecessary. For example, since the NV centers decay and dephase independently of each other, a collective spin description for the decay or dephasing processes is invalid. However, since even the collective coupling λ_{eff} is smaller than the decay rate, there are no collective Rabi-oscillations and the system can safely be described by independent NV centers. This means that it is enough to evaluate the cooling rate for each center and add up all the rates with the appropriate average over the coupling constants. There are many papers in the literature that evaluate mechanical cooling rates using two or multilevel system, which can be adapted for the current problem without the need for a supercomputer. Such an approach would certainly be more appropriate and easier to follow.

Author Reply:

The reviewer observes that our analysis of the proposed cooling protocol is confusing, and he/she claims that a collective spin description is invalid for the decay and dephasing processes because the NV centers within the ensemble decay and dephase independently. In light of the

reviewer comments, we acknowledge that our presentation of this analysis was opaque. We appreciate the reviewer's comments and the opportunity to present our treatment more clearly. We have rewritten portions of the analysis to increase clarity.

We also agree that, because dephasing and decay are stochastic processes, treating the NV center ensemble as a single collective spin is not obviously necessary. Nevertheless, our aim was to present a general treatment that is valid in both the regime of strong coupling (where collective Rabi oscillations can be observed) and weak coupling (where our protocol operates). This treatment facilitates comparisons between different parameter regimes and cooling protocols, and we feel it will be more useful to ourselves and the community. Nonetheless, the referee is correct that the burden of showing validity for the model is on us. In our revised manuscript and SI we address the referee's comments by showing theoretically that by adding up the contributions of individual NV centers to the resonator dynamical equation, the effective ensemble-resonator coupling λ_{eff} naturally emerges. We believe that this new presentation is more intuitive, and we hope that the reviewer agrees that our analysis is valid.

The reviewer also states that there are many papers in the literature that evaluate cooling rates using two or multilevel systems. This is undoubtedly true. For example, our manuscript cites Kepesidis, *et al's* analysis of a single NV center cooling a mechanical resonator via the orbital-strain interaction [1]. In that work, the authors calculate the cooling rate from a three-level NV center coupled to a resonator by working in the Lamb-Dicke (LD) regime wherein the spin dynamics can be adiabatically eliminated from the resonator equation of motion. This LD analysis offers a powerful tool for calculating a protocol's cooling power and can be easily extended to systems with larger numbers of states. In fact, we first approached our proposed protocol with a LD analysis. By calculating the cooling rate from each NV center within the ensemble and summing the rates (as the reviewer suggested), we get the total cooling rate of the ensemble under the LD treatment. For a dense ensemble at room temperature, however, the rates γn_{th} and $\lambda_{eff}\sqrt{\langle n \rangle + 1/2}$ are large enough to put our protocol near the borderline of LD's validity. This motivated our development of the two-state distillation (TSD) treatment, which adapts results already in the literature [2] to solve for the steady state phonon occupancy in the high temperature regime.

To justify our use of the TSD, we compare the accuracy of the LD treatment with the TSD results. We do this by following the procedure described in Supplementary Note 5 and comparing the steady state phonon number predicted by each method with the results of our supercomputer simulations of the seven-level NV-resonator system. As shown in Figs. 1 and 2 below (where the error is taken relative to the supercomputer simulations), the TSD treatment outperforms LD in all cases, and most importantly, as the number of NV centers in the ensemble grows, the error in the TSD treatment grows more slowly than the error in LD. This is because, as λ_{eff} grows, the spin ensemble becomes less of a perturbation on the resonator and the system departs from the Lamb-Dicke regime. More specifically, the approximation $\lambda_{eff}\sqrt{\langle n \rangle + 1/2} \ll T_{2e}^*, T_{1e}$ begins to fail. This surprisingly good performance of the TSD motivated our use of the distilled NV center dynamics in the manuscript. Nevertheless, each method predicts approximately equal cooling power as demonstrated in Table 1 below.

Figure 1: Validation of two-level model for one NV center. Error in the final phonon number predicted by the analytical TSD and LD treatments compared to that predicted by the numerical seven-level simulation for one NV center.

Figure 2: Validation of two-level model for multiple NV centers. (a) Relative error of the analytical TSD and LD treatments with respect to the numerical seven-level simulation for different numbers of NV centers and n_{th} values. (b) Slope of the lines fit to the curves in (a) plotted as a function of n_{th} .

	n_f/n_{th}	
	$\rho = 2.0 \times 10^{18} \text{ cm}^{-3}$	$\rho = 4 \times 10^{20} \text{ cm}^{-3}$
TSD	0.861	0.0300
LD	0.889	0.0389

Table 1: Comparison of different cooling protocol treatments.

Finally, we note that our usage of a supercomputer was only to verify the validity of our distillation to an analytic two-level model. Our analysis of the cooling protocol (Fig. 4d) employed the TSD analysis, and the supercomputer simulations were used only to numerically validate the distillation.

To address this confusion, we have added a concise version of this comparison between TSD and LD to Supplementary Note 5, and we have updated Supplementary Figs. 5 and 6 to include the LD. We agree that a revision was warranted to clarify these points, and we hope that the revised draft avoids confusion on these issues.

- [1] Kepesedis, *et al*, *PRB* **88**, 064105 (2013).
 [2] Wilson-Rae, *et al*, *New J. Phys* **10**, 095007 (2008).

Comment 3:

The cooling rate is reduced by the very low value of $\alpha=0.017$. It seems to me that it should be possible to improve this parameter by a factor of 10 or so by optimizing the driving. Is this true?

Author Reply:

The reviewer notes that the very low value of $\alpha = 0.017$ reduces the cooling rate of the proposed protocol and asks if it is possible to improve α by optimizing the control fields. We agree: this is an important point. We explored this question in detail, and as described in Supplementary Note 8, for the two control fields included in the proposed protocol, α is maximized at ~ 0.017 .

The small value of α can be intuitively understood by comparing the decay rate from the ES to the GS through the metastable singlet state k_{esg} to the rate of decay directly to the GS k_{eg} . Using the rates quoted in the Supplementary Note 3, this gives a ratio of $k_{esg}/k_{eg} \sim 0.02$, which is comparable to $\alpha = 0.017$ and suggests that most of the spin population has become trapped in the metastable singlet $|S_1\rangle$. Examining the steady state density matrix used to calculate α , we see that this is indeed the case. $|S_1\rangle$ contains $\sim 73\%$ of the steady state spin population for the control fields used in the manuscript. Although there has been research into depopulating $|S_1\rangle$ with an additional laser that creates excitations from the metastable state to the conduction band, we felt that invoking this additional protocol was not warranted. Such techniques remain under active experimental study by several groups worldwide and would increase the complexity of the proposed protocol. Without an additional control of the lifetime of $|S_1\rangle$, it is not possible to further increase α .

To address this comment in the main text, we have changed our description of the control fields used to calculate α from “reasonable” to “optimized,” and we have added a concise version of this discussion to Supplementary Note 8.

Comment 4:

The authors use a thin beam theory to evaluate the scalings of the resonator properties like the frequency and coupling constant. However, the resonator dimensions assumed for the evaluation of the cooling rate are no longer thin beams. Since the predicted cooling is very borderline, either a detailed numerical study of the vibrational mode, or 'thinbeam-like' dimensions should be used for estimating the parameters.

Author Reply:

The reviewer points out that we incorrectly apply a thin beam theory to the resonators considered in our analysis of the proposed cooling protocol. We agree that is a fair statement. As argued in the manuscript, the performance of the cooling protocol is insensitive to the physical dimensions of the resonator. The larger dimensions quoted in the original submission were chosen to relax the fabrication requirements needed to implement the proposed protocol. To satisfy the thin beam approximation, we have changed our analysis to treat an appropriately thin beam.

For a fixed resonator frequency, increasing the length-to-thickness ratio reduces the smallest dimension of the proposed resonator. To maintain experimentally viable dimensions, we have modified our analysis to treat a $\omega_m/2\pi = 1$ GHz resonator. Fixing the length-to-thickness ratio at 10, we obtain the reasonable resonator dimensions of $(l, t) = (1.9, 0.19) \mu\text{m}$ for a beam [1,2]. Moreover, we have removed the treatment of cantilevers from our analysis to simplify the discussion of the cooling protocol. Because of our protocol's insensitivity to the device dimensions, this change has no impact on the predicted cooling power or on the main results of the paper.

[1] Khanaliloo, *et al*, *PRX* **5**, 041051 (2015).

[2] Burek, *et al*, *arXiv:1512.04166* (2015).

Comment 5:

In the outlook the authors state that their cooling scheme could be supplemented by other optical cooling schemes to reach the ground state. This is not true (or at least misleading) since the cooling factors will not multiply each other, i.e. if the optical scheme already cools by a factor of 1000 the additional cooling rate provided by the NV centers is negligible.

Author Reply:

The reviewer notes that combining an optomechanical cooling protocol with our proposed protocol will not result in a multiplication of cooling factors. Instead, the two protocols' cooling rates will combine additively. Thus, if the optomechanical protocol cools far more effectively than the NV centers, then the contribution from the NV centers will be negligible (as the reviewer has correctly stated). We agree with this statement.

Our motivation for suggesting the simultaneous implementation of optomechanical cooling was simply to point out that the two approaches are complementary and can work cooperatively to cool a resonator. Optomechanical cooling of the GHz-frequency resonators of interest here has been demonstrated to cool by a factor of $O(100)$ [1,2]. If an optimized NV center ensemble can cool by a factor of ~ 30 on its own, then it could make a meaningful contribution to the total cooling in a simultaneous implementation of the two protocols.

Nevertheless, we understand the reviewer's point that our statement that with simultaneous implementation "cooling a mechanical resonator from room temperature to its zero-point motion may become possible" is misleading. We have removed this sentence from the manuscript and have restructured the paragraph to more appropriately describe the possibility of simultaneous implementation.

[1] Chan, *et al*, *Nature* **89**, 478 (2011).

[2] Safavi-Naeini, *et al*, *PRL* **108**, 033602 (2012).

Reviewer #2:

Manuscript entitled "Cooling a Mechanical Resonator with a Nitrogen-Vacancy Center Ensemble Using a Room Temperature Excited State Spin-Strain Interaction" by MacQuarrie et al. reports an experimental study on the excited state (ES) spin-strain coupling strength and a theoretical proposal and analysis on the use of an ensemble of NV centers to cool a mechanical resonator through the ES spin strain coupling.

The experimental study seems to be solid and provides valuable information on the ES spin-strain coupling. The theoretical proposal is also quite interesting. The subject matter of cooling a mechanical resonator via coupling to an ensemble of defect centers should be of significant interest to both the opto-mechanics and the spin-mechanics communities. My questions and comments here are mostly on the comparison of the proposed ensemble cooling via ES spin-strain coupling with ensemble cooling via ground state (GS) spin-strain coupling and also with ensemble cooling via a phonon-assisted optical transition. These comparisons should further strengthen the paper, making it suitable for publication in Nature Communications.

We thank the reviewer for his/her careful reading and insightful comments on our manuscript. The reviewer identifies the experimental portion of our manuscript as providing "valuable

information on the ES spin-strain coupling” and labels the theoretical section as a “quite interesting” proposal that “should be of significant interest to both the opto-mechanics and the spin-mechanics communities.” The reviewer identifies two alternative mechanisms for coupling NV centers to mechanical resonators and asks us to compare the efficiency of the ES spin-strain interaction with these alternative couplings. We thank the reviewer for these valuable comments and have answered his/her questions below.

Comment 1:

An advantages of using the ES spin-strain coupling instead of GS spin-strain coupling for cooling is the stronger ES coupling strength (about 13.5 times stronger, as shown by the authors). However, I think the figure-of-merit in this case is the cooperativity for a single spin. Although as the authors pointed out, at relatively large densities, the spin decoherence time gets significantly shorter, it is still informative to have a direct comparison of the cooperativity at a moderate NV density. My concern is that T_{-1} time for the ES spin transition is quite short (limited by the excited state lifetime). Without a direct comparison, it is difficult to assess whether cooling due to the ES coupling is indeed much more efficient.

Author Reply:

The reviewer makes the excellent suggestion that the single spin cooperativity is an appropriate figure of merit for accurately comparing a GS spin-strain cooling protocol to the proposed ES cooling protocol. To follow up on this suggestion, we calculate the cooperativity as defined in the manuscript

$$\eta = \lambda^2 T_2^* / \gamma n_{th}.$$

For a fixed mechanical resonator and bath temperature, we can compare the ES and GS cooperativities by calculating the ratio

$$\eta_e / \eta_g = \lambda_e^2 T_{2e}^* / \lambda_g^2 T_{2g}^* = (13.5)^2 T_{2e}^* / T_{2g}^*.$$

To compare the cooperativities at moderate NV center densities, we need to estimate how T_{2g}^* will scale with ρ . In bulk diamond, the GS coherence time is expected to scale with the density of nitrogen impurities n as $T_{2g}^* \sim 1/\beta n$ [1,2]. For a very dense ensemble in a diamond nanostructure, however, effects such as exchange narrowing and the truncation of the spin bath can lead to a narrower linewidth (and a consequently longer T_{2g}^*) than this bulk scaling predicts [3,4]. These effects can be quite large, and NV center ensembles in nanoparticles with densities as high as $1.6 \times 10^{21} \text{ cm}^{-3}$ have been shown to possess ~ 10 MHz linewidths ($T_{2g}^* \sim 16$ ns) [4]. It is currently not well understood how T_{2g}^* will scale with the defect density inside nanostructures, making any predictions of η_e / η_g within a nanostructure speculative.

Nevertheless, we can roughly estimate η_e / η_g by using coherence times measured in bulk diamond. Because the spin-strain interaction couples the $|+1\rangle \leftrightarrow |-1\rangle$ spin transition, the coherence times entering the expression above refer to the coherence of the $\{+1, -1\}$ qubit. As discussed in the manuscript, the ES coherence time is expected to be independent of the NV density, and we take $T_2^e = 6$ ns as reported by [5]. For the GS coherence time, we initially treat an NV center density of $2.8 \times 10^{18} \text{ cm}^{-3}$ for which [2] reported a $\{0, -1\}$ qubit coherence time of $T_{2g}^* = 118$ ns in bulk diamond. If we assume the spin dephasing is dominated by magnetic field noise, this makes the $\{+1, -1\}$ qubit coherence time $T_{2g}^* = 59$ ns. With these numbers, we obtain a cooperativity ratio of $\eta_e / \eta_g = 19$, indicating that the ES spin-strain interaction is a much more efficient route to resonator cooling for this dense ensemble.

For a more moderate NV center density, we turn to the ensemble measured in [6] which had an NV center density of $1.1 \times 10^{14} \text{ cm}^{-3}$ and $T_{2g}^* = 450$ ns for the $\{+1, -1\}$ qubit. This gives a single-

spin cooperativity ratio of $\eta_e/\eta_g = 2.4$, indicating that the ES spin-strain interaction will still provide the more efficient coupling. Moreover, because η_e can be enhanced by a very dense ensemble, the collectively enhanced ES spin-strain interaction offers the most efficient path to resonator cooling.

Finally, we note that the linewidth narrowing effects discussed above suggest that a dense ensemble's ES spin coherence will remain limited by the motional narrowing rate within a nanostructure [3,4]. Our analysis of the proposed cooling protocol is thus expected to remain valid, even at very large defect densities.

We thank the reviewer again for his/her insightful comment. We have added a concise version of this discussion to the main text and an extended version in the SI as Supplementary Note 10.

- [1] Kittel, *et al*, *Phys. Rev.* **90**, 2 (1953).
- [2] Acosta, *et al*, *PRB* **80**, 115202 (2009).
- [3] Smith, *et al*, *Nature* **210**, 692-694 (1966).
- [4] Baranov, *et al*, *Small* **7**, 1533 (2011).
- [5] Fuchs, *et al*, *PRL* **108**, 157602 (2012).
- [6] MacQuarrie, *et al*, *Optica* **2**, 233-238 (2015).

Comment 2:

I understand that the large inhomogeneous broadening of NV optical transitions greatly reduces the effectiveness of using an ensemble of NVs and a phonon-assisted optical transition for cooling purposes. Nevertheless, since the relevant coupling strength through the phonon-assisted optical transition is about four or five orders of magnitude greater than that of the ES spin-strain coupling, it is not clear which one is more efficient. It is probably not realistic to use the NV optical transitions at room temperature. But it is still of significant interest to start the cooling process at low temperature (for example, 80 K or 10 K). In this regard, it will also be useful to make a comparison between ensemble cooling via ES spin-strain coupling and that via phonon-assisted optical transitions under realistic experimental conditions.

Author Reply:

The reviewer raises the interesting point that the NV center's orbital-strain coupling could also be employed to cool a mechanical resonator via phonon-assisted optical transitions. Because this orbital-strain coupling is several orders of magnitude larger than the ES spin-strain coupling, such a cooling protocol could be more efficient than the protocol we propose. As the reviewer points out, such an orbital-strain protocol would have to start from cryogenic temperatures since the NV center ES orbitals dephase above ~ 10 K. This requires a comparison of the two protocols starting at cryogenic temperatures. Unfortunately, the motional narrowing rate that limits the ES spin coherence time T_2^e quenches at ~ 150 K [1,2], effectively erasing the ES spin coherence below this temperature. Because the orbital-strain and ES spin-strain interactions operate in different temperature regimes, it is not possible to make a meaningful comparison between the two interactions.

In response to this comment, we have added a concise version of the above discussion to the main text of the manuscript.

- [1] Batalov, *et al*, *PRL* **102**, 195506 (2009).
- [2] Plakhotnik, *et al*, *PRB* **92**, 081203 (2015).

Reviewer #3:

The manuscript by MacQuarrie et al report new scheme allowing the cooling of mechanical resonator by NV centers in diamond. Cooling of micro and nanomechanical systems is a rapidly growing field of research. Authors of this manuscript show that new approach based on coupling of mechanical motion to spins of NV centers allows very efficient cooling. The novelty of the scheme proposed by MacQuarrie et al is new owing to the use of photoexcited triplet state with an order of magnitude large coupling to mechanical motion of the resonator. The paper consists of proposal and preliminary experiments allowing to quantify the strength of the spin-phonon interaction. The manuscript is well written and from my point of view will be of interest for broad community of scientists. I recommend it for publication after addressing following point.

We thank the reviewer for his/her careful reading of our manuscript. The reviewer notes that our manuscript “will be of interest for a broad community of scientists” and contributes to the “rapidly growing field of research” that studies the cooling of mechanical resonators.

Comment 1:

Authors demonstrate high density ensemble of NVs is capable to cool mechanical mode of the resonator. At the same time populating excited state of NV defect will lead to heating of diamond owing to Stokes shifted emission into phonon sideband. Of course this will lead to a heating of the whole diamond and particular mode of the resonator can be decoupled, there is a limit of temperature of operations of the device owing to decreased lifetime of excited state of NV center at elevated T .

Author Reply:

The reviewer observes that excitation and relaxation through the NV center excited state phonon sideband will heat the diamond, but that the particular mode of the resonator can be decoupled from this heating. Nevertheless, this increase in temperature will affect the NV center spin dynamics through the temperature-dependence of the excited state lifetime. We agree that this is an important consideration, and we thank the reviewer for raising the question. As was demonstrated by [Toyli, *et al*, *PRX* **2**, 031001 (2012).], above ~450 K, thermally-activated non-radiative processes shorten the NV center excited state lifetime T_{1e} . The reviewer asks if the cooling will be limited by this decrease in T_{1e} .

To treat this problem, we estimate the heating a doubly-clamped beam experiences from Stokes emission. Assuming the entire energy of the Stokes phonon $E_s = 59$ meV [1] goes into heat, this heating rate can be approximated by the expression

$$\frac{dQ}{dt} = SNE_s(\Gamma_{opt} + k_{42})$$

where $S = 4$ is the NV center Huang-Rhys factor [2], N is the total number of NV centers (all four defect orientations), Γ_{opt} is the optical pumping rate, and $k_{42} = 65.3$ MHz is the rate of relaxation directly to the GS. Here, we confine the spins to the $\{+1, -1\}$ subspace and ignore the phonons emitted during relaxation through the metastable singlet state because this is a slower process and these phonons are not well understood. This heating is balanced by dissipation into the bulk of the diamond. We model this heat conduction with Fourier’s law according to

$$\frac{dQ}{dt} = -\kappa wt \nabla T$$

where w and t are the width and thickness of the beam, $\kappa = 2000$ Wm⁻¹K⁻¹ is the thermal conductivity of diamond, and $\nabla T = (T(t) - 300 \text{ K})/l$ is the temperature gradient where we take l to be the length of the beam as a conservative estimate. Solving for the steady state of the system, we find the temperature rise

$$\Delta T = T - 300 \text{ K} = \frac{1}{\kappa} S \rho l^2 E_s (\Gamma_{opt} + k_{42})$$

where we have used the NV center density ρ to substitute $N = \rho w l t$. For the parameters used in our manuscript [$\Gamma_{opt} = 130 \text{ MHz}$, $(t, l) = (0.19, 1.9) \mu\text{m}$, $\rho = 7.9 \times 10^{18} \text{ cm}^{-3}$ [3]] and assigning $w = t$, we estimate that Stokes emission will heat the beam by $\Delta T = 52 \text{ mK}$. For the largest defect densities considered in our manuscript ($\rho = 1.6 \times 10^{21} \text{ cm}^{-3}$ [4]), we estimate a temperature rise of $\Delta T = 11 \text{ K}$.

In all cases treated by our manuscript, the temperature rise from Stokes emission is far below the $\sim 150 \text{ K}$ temperature rise needed for thermally-activated pathways to reduce T_{1e} . We thus conclude that heating from Stokes emission will not be a limiting factor in the implementation of our proposed cooling protocol.

- [1] Doherty, *et al*, *Physics Reports* **528**, 1-45 (2013).
- [2] Acosta, *et al*, *PRB* **80**, 115202 (2009).
- [3] Choi, *et al*, *arXiv:1608.05471* (2016).
- [4] Baranov, *et al*, *Small* **7**, 1533 (2011).

REVIEWERS' COMMENTS:

Reviewer #1 (Remarks to the Author):

In the resubmitted version the authors have substantially revised the manuscript and addressed in detail all the points that I have raised in my previous report. In particular, the derivation of the cooling rate is now much easier to follow and it is shown that different approaches for calculating the final occupation number lead to very similar predictions. With revised analysis it is now much more apparent that the predicted cooling rates are correct and physically sensible. I really appreciate the effort that the authors have put into this revision and think that the manuscript has improved a lot.

In summary, while I still think that conditions under which the proposed cooling scheme works are quite "extreme", they are not completely unrealistic and partially supported by previous measurements. In addition, the first part of the paper reports on the first measurement of the excited state strain coupling, which apart from cooling applications, could be relevant for other type of strain-based manipulation schemes, etc. I have no further complaints and I can recommend the revised manuscript for publications.

Reviewer #2 (Remarks to the Author):

I thank the authors for the detailed response. I am satisfied with the response and recommend the publication of the manuscript in Nature Communication.

I have one minor comment with regard to the very high NV density ($>10^{18}/\text{cm}^3$) used in some of the estimates. Such a high defect density can lead to strong degradation of the mechanical Q-factor.

Reviewer #3 (Remarks to the Author):

All comments were addressed. I recommend the manuscript for publication in the present form.

Cooling a Mechanical Resonator with a Nitrogen-Vacancy Center Ensemble Using a Room Temperature Excited State Spin-Strain Interaction:
Response to Reviewers

E. R. MacQuarrie, M. Otten, S. K. Gray, and G. D. Fuchs

Reviewer #1:

In the resubmitted version the authors have substantially revised the manuscript and addressed in detail all the points that I have raised in my previous report. In particular, the derivation of the cooling rate is now much easier to follow and it is shown that different approaches for calculating the final occupation number lead to very similar predictions. With revised analysis it is now much more apparent that the predicted cooling rates are correct and physically sensible. I really appreciate the effort that the authors have put into this revision and think that the manuscript has improved a lot.

In summary, while I still think that conditions under which the proposed cooling scheme works are quite "extreme", they are not completely unrealistic and partially supported by previous measurements. In addition, the first part of the paper reports on the first measurement of the excited state strain coupling, which apart from cooling applications, could be relevant for other type of strain-based manipulation schemes, etc. I have no further complaints and I can recommend the revised manuscript for publications.

Author Reply:

We would like to thank the reviewer for his/her careful reading of our revised manuscript. The reviewer notes that our predictions for the cooling protocol performance are "correct and physically sensible" and that our measurement of the excited state spin-strain coupling "could be relevant for other types of strain-based manipulation schemes." We agree that the suggested revisions have improved our manuscript and we appreciate the reviewer's recommendation that our manuscript be accepted for publication.

Reviewer #2:

I thank the authors for the detailed response. I am satisfied with the response and recommend the publication of the manuscript in Nature Communication.

I have one minor comment with regard to the very high NV density ($>10^{18}/\text{cm}^3$) used in some of the estimates. Such a high defect density can lead to strong degradation of the mechanical Q-factor.

Author Reply:

The reviewer makes the minor comment that the very high NV center density used in our predications of the cooling performance can degrade the Q-factor of the mechanical resonator. We agree that this is a concern and already note this possibility in the main text of the manuscript. Nevertheless, the degradation of the Q-factor with increasing NV center density has not been studied in the gigahertz-frequency resonators of interest here. We thus present predictions for resonators with the ideal Q-factor and motivate future experiments to determine how a large NV center density will affect the Q-factor of our proposed resonators.

We are glad that our response has satisfied the reviewer's concerns and appreciate the reviewer's recommendation that our manuscript be accepted for publication.

Reviewer #3:

All comments were addressed. I recommend the manuscript for publication in the present form.

Author Reply:

We are glad that our response has satisfied the reviewer's concerns and appreciate the reviewer's recommendation that our manuscript be accepted for publication.